# Internal Reference Gene Selection under Different Hormone Stresses in Multipurpose Timber Yielding Tree *Neolamarckia cadamba*

**Deng Zhang** [1,2,3]**, Jingjian Li** [1,2,3]**, Buye Li** [1,2,3]**, Chunmei Li** [1,2,3]**, Xiaoyang Chen** [1,3,]*
**and Kunxi Ouyang** [2,]*

1   State Key Laboratory for Conservaion and Utilization of Subtropical Agro-Bioresources,
    South China Agricultural University, Wushan Road 483, Tianhe District, Guangzhou 510642, China;
    zhangdeng1992@gmail.com (D.Z.); calljoneli@gmail.com (J.L.); byl766798@gmail.com (B.L.);
    dachunli134@gmail.com (C.L.)
2   Guangdong Key Laboratory for Innovative Development and Utilization of Forest Plant Germplasm,
    South China Agricultural University, Wushan Road 483, Tianhe District, Guangzhou 510642, China
3   Guangdong Province Research Center of Woody Forage Engineering Technology,
    College of Forestry and Landscape Architecture, South China Agricultural University,
    Wushan Road 483, Tianhe District, Guangzhou 510642, China
*   Correspondence: xychen@scau.edu.cn (X.C.); kxouyang@scau.edu.cn (K.O.)

**Abstract:** *Neolamarckia cadamba*, a member of the Rubiaceae family, is widely distributed throughout South Asia and South China. In order to acquire reliable and repeatable results, the use of a suitable internal reference gene to normalize the RT-qPCR data is essential. In this study, we reported the validation of housekeeping genes to identify the most suitable internal reference gene(s) for normalization of qPCR data obtained among different tissues (bud, leaf, cambium region) under different hormone stresses. Here, ΔCt, geNorm, NormFinder, and BestKeeper analyses were carried out to analyze the normalization of qPCR data of twenty-one reference gene families (*ACT, CAC, CYP, EF1α, eIF, FPS1, FBK, GAPDH, RAN, PEPKR1, PP2A, RPL, RPS, RuBP, SAMDC, TEF, Tub-α, Tub-β, UBCE, UBQ, UPL*) including 43 genes. The results showed that *FPS1, RPL,* and *FBK* were the most stable reference genes across all of the tested samples. In addition, the expression of *NcEXPA8*, one gene of interest that plays an important role in regulating cell wall extension, under different phytohormone stresses was used to further confirm the validated reference genes. Taken together, our results provide guidelines for reference gene selection under different phytohormone stresses and a foundation for more accurate and widespread use of RT-qPCR in *N. cadamba*.

**Keywords:** *Neolamarckia cadamba*; RT-qPCR; reference genes; hormone stresses

## 1. Introduction

Real-time quantitative PCR (RT-qPCR) is the preferred method for the validation of high-throughput or microarray results and for determining gene expression levels due to its good reproducibility, high sensitivity, accurate quantitation, and fast response [1–3]. Several factors have impacts on the experimental results in gene expression analysis, such as the initial template amounts among different tissue cells, RNA quality, and enzymatic efficiencies [4], although RT-qPCR is widely used to qualify mRNA levels during biological changes. To avoid severe pitfalls and biases in data analysis, a number of strategies have been proposed to normalize RT-qPCR data. However, so far, reference genes that should be expressed at a constant level across various conditions,

such as developmental stages or tissue types, are the most commonly used to normalize RT-qPCR data and to control the experimental possible errors generated in the quantification of gene expressions [5].

The housekeeping genes most commonly considered and used as internal controls include 18S ribosomal RNA (18S rRNA), glyceraldehyde-3-phosphate dehydrogenase (GAPDH), elongation factor (EF), ubiquitin-binding enzyme (UBCE), alpha tubulin (Tub-α), and β-tubulin (Tub-β) [6]. However, so far, several reports have demonstrated that there are no "universal" reference genes with invariant expression [7–10]. Since RT-qPCR is a highly sensitive tool, choosing unstable reference genes with large fluctuations in expression will lead to errors or even opposite conclusions in biological data interpretation [11]. Thus, in order to acquire reliable and repeatable results, the selection and systematic validation of suitable reference genes as internal controls is an essential prerequisite in RT-qPCR normalization for every specific experimental condition [5]. Meanwhile, several algorithms, such as geNorm [12], NormFinder [13], BestKeeper [14], and ΔCt [15], have been well developed to validate the most stable reference gene(s) from a series of candidate genes under a given set of experimental conditions. Recently, RefFinder [16] has been developed as a comprehensive evaluation platform, which can integrate the four algorithms above and rank the overall stability.

*N. cadamba*, a member of the Rubiaceae family, is widely distributed throughout South Asia and South China [17]. It has received high praise in the Philippines, where it has been described as "a gem of a tree", and was universally accepted as "a miraculous tree" at the World Forestry Congress in 1972 because of its fast growth [18]. In addition, the extractions from different tissues of *N. cadamba* contain secondary metabolites, such as phenolic compounds, flavonoids, alkaloids, and others, which have been used in the treatment of eye infections, skin diseases, indigestion, bleeding gums, stomatitis, cough, fever, anemia, and stomach aches [19–21]. Changes in hormone concentration or sensitivity, which can be triggered under biotic and abiotic stress conditions, mediate a series of plant adaptive responses [22]. Furthermore, the importance of abscisic acid (ABA), gibberellic acid (GA₃), auxin (IAA), 6-benzylaminopurine (6-BA), brassinosteroids (BRs), methyl jasmonate (MeJA), and ethylene (ET) as primary signals in the regulation of the plant's stress responses has been well revealed [23–28]. Although stable internal reference genes have been obtained among different tissues of *N. cadamba* [7], the stable genes for different hormone treatments are still unclear.

In the present study, we report the validation of housekeeping genes to identify the most suitable internal reference genes for normalization of qPCR data obtained among different tissues (bud, leaf, cambium region) of *N. cadamba* under different hormone stresses (ABA, GA₃, IAA, 6-BA, BRs, MeJA and ETH). Here, ΔCt, geNorm, NormFinder, BestKeeper, and RefFinder algorithms are used to analyze normalization of qPCR data of 43 candidate reference genes belonging to 21 gene families (*ACT, CAC, CYP, EF1α, eIF, FPS1, FBK, GAPDH, RAN, PEPKR1, PP2A, RPL, RPS, RuBP, SAMDC, TEF, Tub-α, Tub-β, UBCE, UBQ, UPL*). Additionally, to illustrate the usefulness of the new reference genes, the expression analysis of *NcEXPA8*, one gene of interest playing an important role in regulating cell wall extension, under different phytohormone stresses is presented. Our data provide a superior set of validated internal reference genes that are stable in different tissues of *N. cadamba* under phytohormone stresses for the expression analysis of important target genes.

## 2. Materials and Methods

### 2.1. Plant Materials

Tissue culture seedlings of *N. cadamba* were grown in a greenhouse under standard conditions (16 h day at 25 °C, 8 h night at 22 °C) to a height of 50 cm. The seedlings were sprayed with 10 mg/L ABA, 10 mg/L GA₃, 10 mg/L IAA, 10 mg/L 6-BA, 10 mg/L ethephon (ETH), 100 μM MeJA, or 0.1 mg/L BR and were immediately covered with transparent plastic bags after ETH or MeJA spraying. The buds, cambium regions, and leaves were sampled at 0, 3, 6, 12, 24, and 48 h after hormone stresses. The cambium regions were collected according to a previously described method [29]. Each tissue

was collected from three individual plants representing three biological replicates. All samples were immediately frozen in liquid nitrogen and stored at −80 °C in a refrigerator.

## 2.2. Identification of Candidate Internal Reference Genes

The amino acid sequences of a total of 23 housekeeping gene families (*ACT, APT, CAC, CYP, EF1α, eIF, FPS1, FBK, GAPDH, RAN, PEPKR1, PP2A, RBCL, RPL, RPS, RuBP, SAMDC, TEF, Tub-α, Tub-β, UBCE, UBQ, UPL*) of *Arabidopsis thaliana* were downloaded from the *A. thaliana* TAIR10 database. UniGenes from a RNA-seq project of *N. cadamba* (http://www.ncbi.nlm.nih.gov/bioproject/PRJNA232616) [30] were searched against this database using the local NCBI-2.2.30 + BLASTx algorithm (*E*-value ≤ $1 \times 10^{-10}$). The UniGene sequences were double-checked by BLASTx searches against protein databases, including the NCBI non-redundant (nr) database and the *A. thaliana* TAIR10 database.

## 2.3. Total RNA Isolation

Total RNA from each sample was isolated using CTAB plus the OMEGA Plant RNA isolation kit, as described previously [29]. Frozen tissue samples were ground to a fine powder in liquid nitrogen using a mortar and pestle. Then, 100 mg samples of the powder were transferred into individual 1.5 mL RNase-free tubes containing 600 mL of prewarmed extraction buffer at 60 °C. The extraction buffer contained the following: 2% CTAB, 2% polyvinylpyrrolidone (PVP) K-40, 100 mmol/L TrisHCl (pH 8.0), 25 mmol/L ethylenediaminetetraacetic acid (EDTA; pH 8.0), 2.0 mol/L NaCl, 2 g/L spermidine, and 2% b-mercaptoethanol (added immediately before use). The extracts were mixed by vortexing and incubated at 60 °C in a water bath for 10 min with several rounds of vigorous shaking. An equal volume of chloroform/isoamyl alcohol (24:1) was added to the homogenate and was mixed completely by vortexing. The mixtures were centrifuged at 12,000 r/min for 10 min at 4 °C, except for bud samples, which were centrifuged for 20 min. The supernatant was transferred to a new tube and the above step was repeated. The supernatant was then transferred to a new tube and the next operation was carried out according to the instructions of the Plant RNA Kit (R6827, OMEGA). Finally, RNA was eluted with 40 μL of DEPC $H_2O$.

## 2.4. cDNA Synthesis and Real-Time RT-PCR

Total RNA (0.5 μg) was reverse transcribed into the first strand cDNA according to the PrimeScript RT Master Mix kit (TaKaRa, Guangzhou, China) instructions. The single-stranded cDNA was diluted 15-fold for PCR amplification and the amplified products were examined by 2% agarose gel electrophoresis. The PCR total volume of 25 μL contained 12.5 μL 2 × Ex Taq Master Mix, 1 μL Primer F (5 μmol·$L^{-1}$), 1 μL Primer R (5 μmol·$L^{-1}$), 2 μL cDNA, 8.5 μL ddH$_2$O. The reaction procedure was as follows: 94 °C for 3 min, 35 cycles (94 °C for 30 s, 58 °C for 30 s, 72 °C for 15 s), 72 °C for 10 min, 10 °C for storage.

RT-qPCR amplification was performed in 96-well plates on an LC480 instrument (Roche, CA, USA) using SYBR® Premix Ex Taq™ (TaKaRa, Guangzhou, China). The real-time PCR volume of 10 μL contained 5 μL 2xSYBR Premix Ex Taq II, 0.5 μL primer F (5 μmol·$L^{-1}$), 0.5 μL primer R (5 μmol·$L^{-1}$), 2 μL cDNA, and 2 μL ddH$_2$O. The thermocycling conditions were as follows: 95 °C for 30 s, 40 cycles (95 °C for 5 s, 56 °C for 30 s, 72 °C for 30 s), 72 °C for 2 min, and an infinite hold at 10 °C. The melting curves ranging from 56 °C to 95 °C were evaluated in each reaction to check the specificity of the amplicons after the final PCR cycle.

## 2.5. Data Analysis

Standard curves were generated in Microsoft Excel 2016 to calculate the gene-specific PCR efficiency and the correlation coefficient from 10-fold series dilution of a mixed cDNA (bud, leaf, cambium region) template for each primer pair. The PCR amplification efficiency (E) and the correlation coefficient were calculated using the slope of the standard curve according to equation $E = (10^{-1/\text{slope}} − 1) \times 100$ [10].

The stability of the 43 candidate reference genes was evaluated by four algorithms—geNorm [12], NormFinder [13], BestKeeper [14], and the ΔCt [15] method. Finally, RefFinder [16], a comprehensive evaluation platform integrating the four algorithms above, ranked the overall stabilities of these 43 candidate genes. Pairwise variations based on the geNorm calculation were used to determine the optimal number of candidate reference genes for accurate normalization

*2.6. Validation of Internal Reference Genes*

Expansins are a class of specific proteins with plant cell wall extension ability [31]. Plant hormones and external stimuli (such as light, drought, hormones, salt stress, and hypoxia) affect the expression of expansin genes [32]. Therefore, in order to detect differences among different internal reference genes used in data normalization, the relative expression of *NcEXPA8* [17] in the buds, leaves, and cambium regions of *N cadamba* under different hormone stresses was evaluated. According to the results of RefFinder analysis, the most stable reference gene (combination) and the most unstable reference gene were selected as internal controls, respectively.

## 3. Results

*3.1. Primer Quality and CT Analysis of Candidate Internal Reference Genes*

A total of 415 *N. cadamba* UniGenes with high sequence identity (*E*-value $\leq 1 \times 10^{-10}$) corresponding to 23 *A. thaliana* housekeeping gene families downloaded from the TAIR10 database were found in the stem database (Supplementary S1). Primer5 (Premier Biosoft Interpairs, Palo Alto, CA, USA) was used to design primers for these genes, but there were no suitable primers for *APT* or *RBCL* gene families. Therefore, the 43 candidate internal reference genes belonging to the other 21 gene families were selected and primers were designed for them (Table 1). The cDNA of the leaf tissue was used as the template for the amplification of each candidate internal reference gene fragment by PCR. The amplified product with a single band was detected by 2% agarose gel electrophoresis and its fragment size was consistent with predicted value (Figure 1A). Furthermore, the fragment was confirmed to be correct by sequencing. These primer pairs were further checked by RT-qPCR and the melting curve of the amplicon from each primer pair had only one signal peak (Figure 1B), indicating high specificity. The amplification efficiency of the primers ranged from 96.1% to 105.8%. All of these factors suggested that these primers were appropriate and could provide reliable results in RT-qPCR.

Figure 2 shows the distribution range of the cycle threshold (CT) values of 43 candidate internal reference genes selected in this study under different hormone stresses. The CT value for each gene was taken from 324 samples (6 time-points per tissue set and 3 tissue sets per hormone stress). The mean CT values of internal reference genes ranged 19.02–30.07. Here, g27 is the most expressed internal reference gene (mean CT = 19.02), but g8 had the lowest expression level (mean CT = 30.07). Additionally, with regard to individual reference gene expression variation across all studied RNA samples, all of the reference genes had high expression variation (above 7 cycles). Furthermore, 10 of them showed lower expression variation (below 10 cycles), including g4, g10–14, g21, g23–24, and g26. However, the other genes had much higher expression variations (above 10 cycles). The wide expression ranges of the 43 tested reference genes indicated that none of the selected genes had constant expression in different *N cadamba* samples. Therefore, it was extremely important to evaluate and select a reliable reference gene for gene expression normalization under a certain condition in *N cadamba*.

**Table 1.** Selected candidate reference genes, primers, and amplicon characteristics.

| Gene Name | UniGene ID | Reference Gene ID | F/R Primer (5′—3′) | Amplicator Length (bp) | Efficiency (%) | Correlation Coefficient ($R^2$) |
|---|---|---|---|---|---|---|
| *ACT* | comp52737_c0 | g1[a] | ATGTTGAAGCCTGTTCCATTGT TAACTAATAACAGAAGCATTCATCCA | 114 | 97.6 | 0.997 |
| | comp79635_c0 | g2 | CTTCTGAGGTTATGGAGCAATCT CGATAAATCAAAACTTCAAGCC | 101 | 105 | 0.993 |
| *CAC* | comp48976_c0 | g3 | CTCAGAGAACGCTGCTGACTAC GAGCCAAGGGAAACAAGATAA | 161 | 104.8 | 0.996 |
| *CYP* | comp67418_c0 | g4 | GGGGTCTCACGCTCTTTACT GGATTGGATTGGGTTGGTT | 83 | 96.8 | 0.993 |
| | comp75463_c0 | g5 | CCCCAGCAAGAAGACCACT TTGACCATGAATCCCAACCA | 213 | 102.1 | 0.996 |
| | comp77969_c0 | g6 | ATAGCATCCCAACCGAACA CCCTCTTGCCTCCTGTGTAT | 187 | 102.1 | 0.997 |
| *EF1α* | comp87079_c1 | g7 | ACCAGCATCACCGTTCTTCA GTCCTCGATTGCCACACCT | 123 | 98 | 0.996 |
| | comp87526_c0 | g8 | AATCAGACAGAAACCCCTCAA GAACCTCTCAATCACACGCTT | 245 | 101.8 | 0.994 |
| *eIF* | comp6386_c0 | g9 | GTTGAAACTTCTTGGACATCG CTTGAGACACTGATTTGTATGAGA | 250 | 103.3 | 0.991 |
| *FPS1* | comp72548_c0 | g10 | TGATAATCTGGCTTCCACCTT TGGGAGGAACTCAATCTCCTAC | 112 | 103.6 | 0.992 |
| | comp75377_c0 | g11 | TATCAGGCTCAGCATTCCACT TTGCCACAATAACACATCCAT | 212 | 101.6 | 0.994 |
| *FBK* | comp78454_c0 | g12 | AAGGCCAATTCTGTTCAAGC CCTAGAGGGAAAGACATGACTG | 143 | 96.1 | 0.997 |
| | comp78817_c0 | g13 | GCAAACGGGGTAAAAGGA AAAGGGTAAGAGTGACGACAGC | 102 | 99.6 | 0.993 |
| *GAPDH* | comp78593_c0 | g14 | TGTTCCAAGTGGGCATTTAC CGCTCTGAGGTGTTAATAAGTG | 247 | 104.3 | 0.99 |
| | comp80828_c1 | g15 | CTGAGCATTTTTTAGGCTTGTC TCAGATTCATGTGGCAGTCG | 151 | 103.5 | 0.992 |

**Table 1.** *Cont.*

| Gene Name | UniGene ID | Reference Gene ID | F/R Primer (5′—3′) | Amplicator Length (bp) | Efficiency (%) | Correlation Coefficient ($R^2$) |
|---|---|---|---|---|---|---|
| *RAN* | comp85262_c0 | g16 | TCTCGCAACCTGCCTCTT TATCACTCCCATCTTCGCAC | 257 | 101.2 | 0.99 |
| *PEPKR1* | comp75525_c0 | g17 | CGACCTCACATTCCTCATTAC ACATAGACCATCCAGAGCCCA | 291 | 97.7 | 0.995 |
| | comp80613_c0 | g18 | TACATAGACCATCCAGAGCCA GCAAAAGGGCAAGCAACAG | 112 | 102.4 | 0.991 |
| *PP2A* | comp81334_c1 | g19 | CTGGGTGGGAAAGATGTG CTTGGGCAATAGGCTGAC | 142 | 104.6 | 0.995 |
| | comp52412_c0 | g20 | ATGTTGGATGATATTAGTGGTGTG TCATAGGAAAATAGACCTCTGGTT | 161 | 100.3 | 0.992 |
| *RPL* | comp46755_c0 | g21 | CTGAGGATTGTTAGCAGTTGAC ACCAGAAAACAGACCACCTAAG | 119 | 103.4 | 0.993 |
| | comp52434_c0 | g22 | AAGGAAGGTAAAGCAGGGAA GCATGGGCAGGGATATAAAC | 177 | 98.4 | 0.995 |
| | comp87976_c0 | g23 | CACGCAGCATAGCCAAAC AGGCAGTTCTCTGATTCTTTTG | 157 | 104.5 | 0.991 |
| *RPS* | comp65909_c1 | g24 | GCTATGGTAGTCTCCCGAAAG GGGGGAACAAGACTAAGGGT | 182 | 104.4 | 0.992 |
| | comp67276_c0 | g25 | TTTTGTTTCCCCTCTTTGC AACCTTGAACAACCTGTGTAGAA | 97 | 102.6 | 0.991 |
| | comp71526_c0 | g26 | CGGTTACACAAGGTTGAATGA AGAGGGTCTGGATTTGAGTGA | 117 | 104.5 | 0.996 |
| *RuBP* | comp47386_c0 | g27 | CAGCACCGTAATCCATAAAAC CAAGCAGCCCAGCAAGTC | 226 | 104.6 | 0.993 |
| | comp88001_c0 | g28 | ACAGGATGGGTAGAAAGAGGC AGGATTGAGCCGAATACAACG | 210 | 104.9 | 0.996 |
| *SAMDC* | comp44802_c0 | g29 | TCTTCGTGGCACTTCTCTCC ACAGGGTGTTGACTTGTTTCC | 133 | 101.5 | 0.993 |
| | comp71874_c0 | g30 | ATAAGGTCTCTTCTTGTTCGTGTAG GACTGAACAGCAACAGGAATAAT | 178 | 103.5 | 0.994 |
| | comp80075_c0 | g31 | GCTGCCTGTGGGTCTCCTA GTAAACCCCAATGCTACTCCT | 85 | 104.8 | 0.998 |

**Table 1.** *Cont.*

| Gene Name | UniGene ID | Reference Gene ID | F/R Primer (5'—3') | Amplicator Length (bp) | Efficiency (%) | Correlation Coefficient ($R^2$) |
|---|---|---|---|---|---|---|
| *TEF* | comp65909_c1 | g32 | GCTATGGTAGTCTCCCGAAAG CTGGGGGAACAAGACTAAGG | 184 | 100.5 | 0.995 |
|  | comp70791_c0 | g33 | TCAACCAACCGTTCCTACC ACAACAGTCCTTTGCCACC | 195 | 105.2 | 0.99 |
| *Tub-α* | comp70323_c2 | g34 | GGTGGTGGAACTGGCTCTG GGCAAATGTCATAGATGGCTT | 217 | 103.3 | 0.993 |
|  | comp76448_c4 | g35 | AAGGAGGGAATGAGTGGAG ACTATGGCAAGAAGTCAAAGC | 107 | 103.4 | 0.99 |
| *Tub-β* | comp66056_c0 | g36 | GCAAGAAAGCCTTCCTCCTAA TTCCCAACAATGTCAAATCAA | 153 | 103.9 | 0.999 |
|  | comp79707_c1 | g37 | TTCAGGAGAGTCAGCGAGC CATCGTCTTCATATTCCCCTT | 187 | 100.4 | 0.999 |
| *UBCE* | comp79182_c1 | g38 | TCCTTGCTTGTGGCGTCA CACGGGTGTCAAATCTGGC | 213 | 105.8 | 0.999 |
| *UBQ* | comp67366_c0 | g39 | GACGGGAGGACCTTAGCA CTCGGAGACGGAGAACAA | 298 | 105.5 | 0.993 |
|  | comp82561_c0 | g40 | GCATTTGTGTCTTGCCTCTTTAT GCGATGAGCAACATTCCTTTA | 186 | 105.5 | 0.993 |
|  | comp75872_c1 | g41 | TCTTGAAGGGAATGGTGTTTTG AGATGTTAGGAGGACTGAGGAT | 267 | 105.3 | 0.993 |
| *UPL* | comp87122_c0 | g42 | GGTTGGTGGTAGAGTTGTGACTC CGAGCACTACCACGACACG | 182 | 105.8 | 0.995 |
|  | comp88840_c0 | g43 | CTGCTCGTTGGTATGTAATGG TCAGGCAATCCAAAGACAACT | 128 | 104.3 | 0.99 |

[a] Reference gene ID for each candidate reference gene.

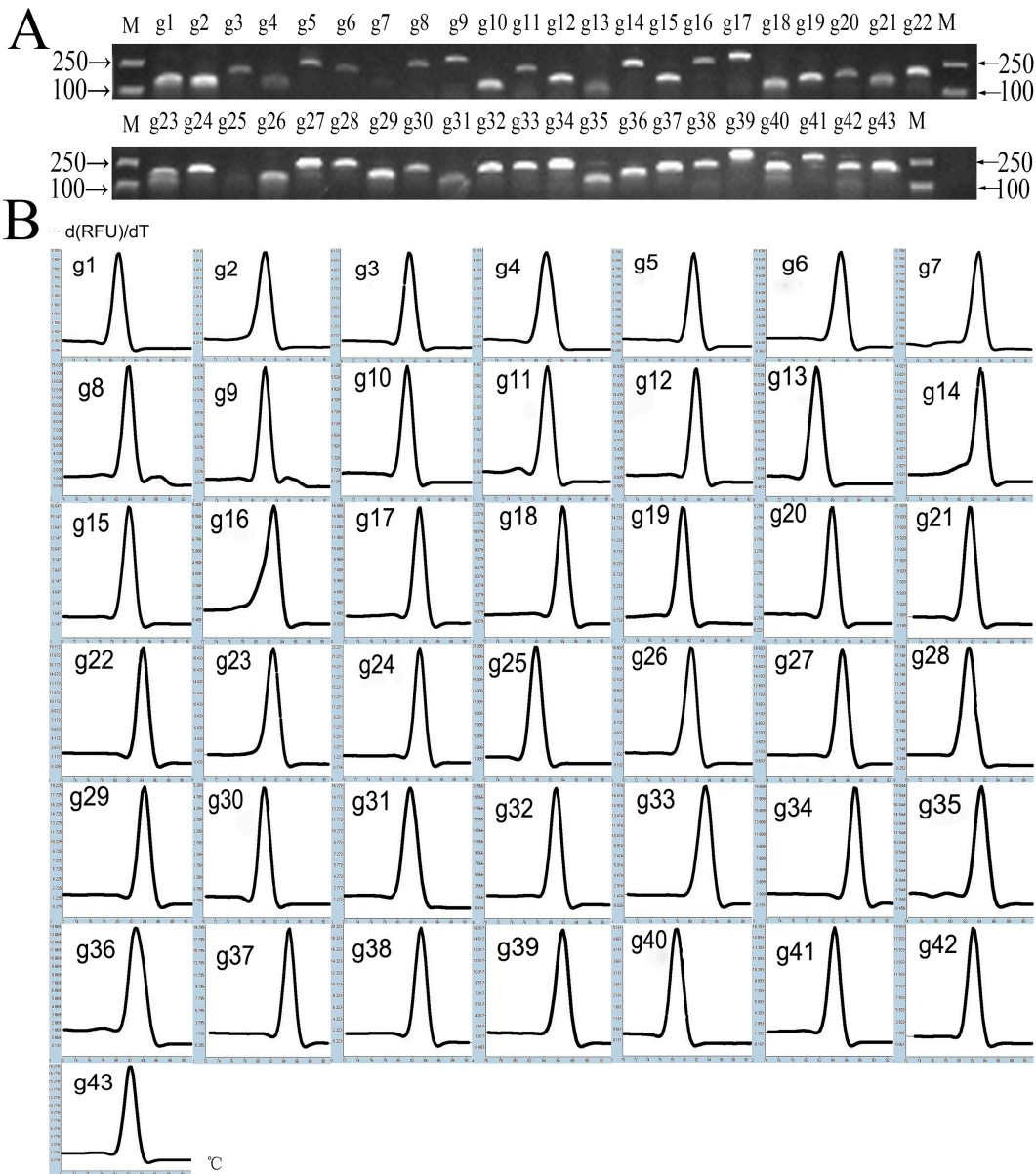

**Figure 1.** Specificity of primer pairs for RT-qPCR amplification. (**A**) Agarose gel (2%) showing amplification of a specific PCR product of an expected size for each candidate reference gene tested in the study. (**B**) Melting curves for the 43 candidate reference genes with single peaks.

### 3.2. Analysis of Gene Expression Stability

The expression stability of the 43 candidate reference genes was determined using geNorm, NormFinder, ΔCt, and BestKeeper, and ranked by RefFinder. Table 2 shows the differences in the stable internal reference genes among different tissues under different hormone treatments. The detailed results for each tissue and different tissues under each hormone stress are shown in the Tables S1–S28 (Supplementary S2). Furthermore, the gene expression stability was analyzed for the same tissues under different hormone stresses. RefFinder's comprehensive ranking results showed that g10, g23, and g38 was the most stable reference genes in the buds, leaves, and cambium regions, respectively, but g17 was the most unstable in the buds and g9 was the most unstable in the leaves and cambium regions (Tables S29–S32; Supplementary S2). Finally, the CT values of all samples were analyzed together to find the universal reference gene under all hormones stress conditions. The RefFinder results showed that g10 was the most stable reference gene and the most unstable was g17.

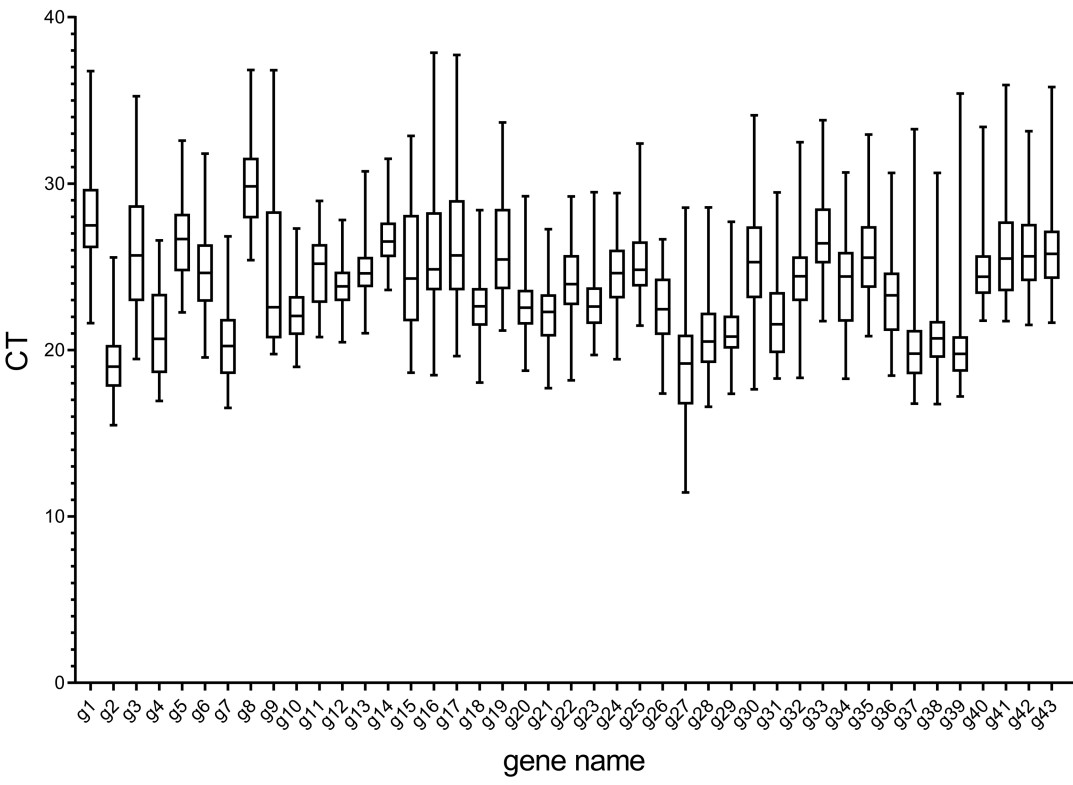

**Figure 2.** RT-qPCR CT values for the candidate reference genes. Expression data displayed as CT values for each candidate reference gene in all *N cadamba* samples. The line across the box is depicted as the median. The box indicates the 25th and 75th percentiles. Whiskers represent the maximum and minimum values.

**Table 2.** Summary of the most and least stable internal reference genes with different treatments.

| Treatments | Buds (Most Stable/Least Stable) | Leaves (Most Stable/Least Stable) | Cambium Region (Most Stable/Least Stable) | Total (Most Stable/Least Stable) |
|---|---|---|---|---|
| GA$_3$ | g29/g16 | g13/g16 | g38/g3 | g20/g16 |
| ETH | g39/g3 | g38/g17 | g20/g15 | g20/g17 |
| BR | g42/g30 | g25/g8 | g22/g19 | g9/g27 |
| 6-BA | g20/g16 | g20/g16 | g12/g16 | g20/g39 |
| MeJA | g37/g9 | g38/g9 | g20/g9 | g38/g9 |
| ABA | g10/g9 | g23/g9 | g20/g9 | g38/g9 |
| IAA | g38/g18 | g7/g18 | g12/g18 | g7/g27 |
| Total (most stable/least stable) | g10/g17 | g23/g9 | g38/g9 | g10/g17 |

### 3.3. Optimization of the Number of Reference Genes Required for RT-qPCR Analysis

It is also important to know the optimum number of reference genes that are required to normalize RT-qPCR data for the given samples in an experiment [8]. In addition, geNorm also calculated the paired variation value ($V_{n/n+1}$) of standardized factors after the introduction of one new reference gene and determined the optimum number of reference genes based on this ratio. The default $V_{n/n+1}$ threshold value of the software is 0.15, below which there is no necessary inclusion of an additional reference gene [12]. The experimental results showed the number of reference genes needed for RT-qPCR data normalization for the different sample sets under hormone stresses (Figure 3). All leaf samples under different hormone stresses or all samples under 6-BA stress needed 7 reference genes for RT-qPCR data normalization. For all samples under GA$_3$ stress, all bud samples, or all cambium

region samples under different hormone stresses, 5 internal reference genes should be used. Under BR or ABA stress, 4 internal reference genes should be used in all samples. For all samples under the stress of ETH or IAA, or all leaf samples under 6-BA or ABA stress, the optimum number of internal reference genes was 3. In the other sample sets, only two reference genes would be sufficient, since the $V_{2/3}$ values in these sample sets were inferior to the 0.15 cut-off level.

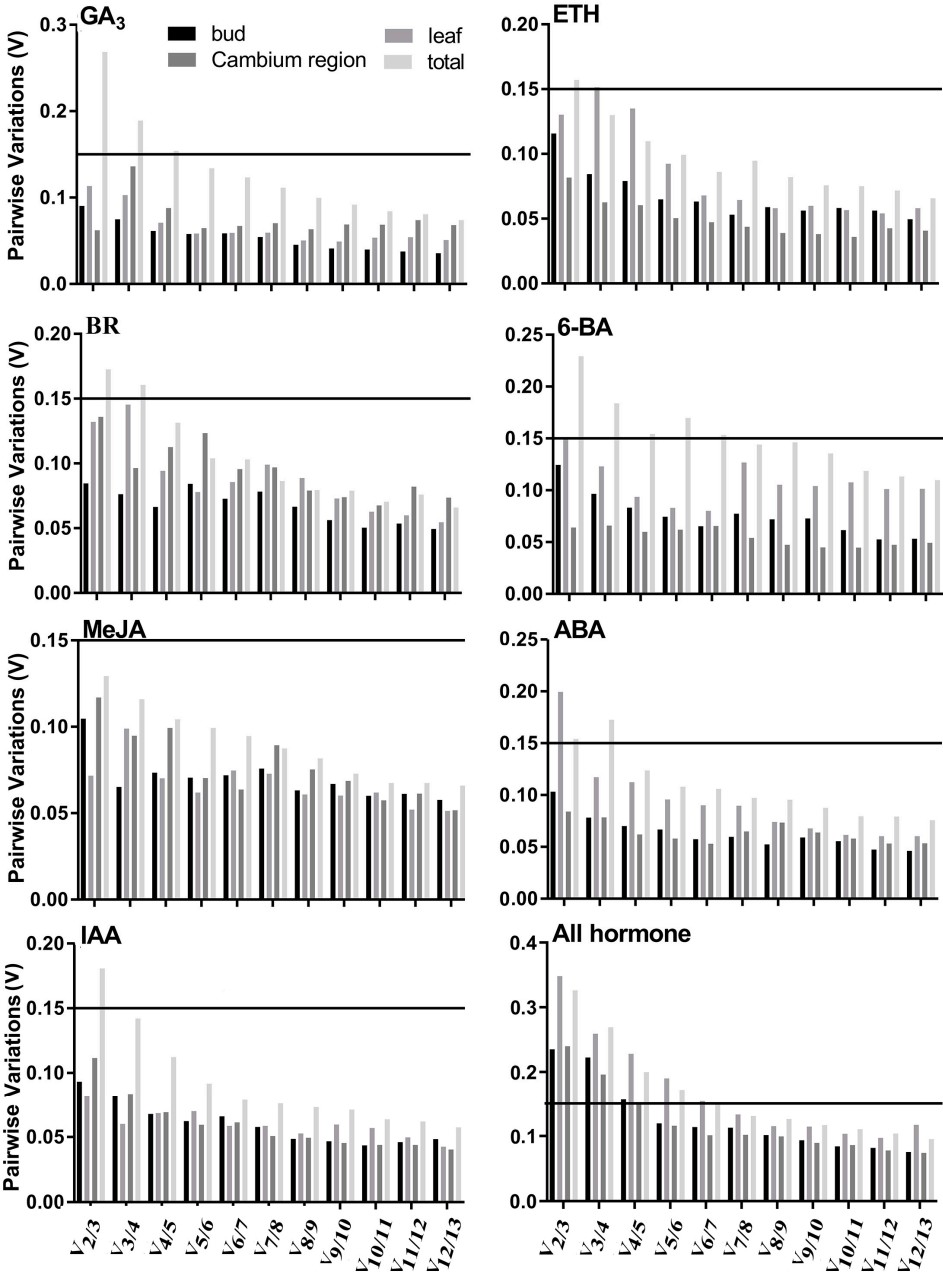

**Figure 3.** Pairwise variation (V) of the candidate gene was calculated by geNorm in each sample set. $V_{n/n+1}$ was used to ascertain the optimum number of reference genes. The ordinate value of the solid lines is the default $V_{n/n+1}$ threshold value of 0.15.

When we considered the outcomes of the four algorithms, all analyses did not produce consistent results (Tables S1–S32 of Supplementary S2). RefFinder can integrate the rankings of the four algorithms and rank these from the most stable to the least stable based on the calculation of the geometric mean of the four algorithms—the smaller the geometric mean, the greater the stability of the reference gene expression. It is not practical for more than three reference genes to be used together in RT-qPCR

under a given experimental condition. However, more than three reference genes would be used together in RT-qPCR for several sample sets according to pairwise variation analysis by geNorm in the present study (Figure 3). Therefore, to obtain the number of suitable reference genes used under given experimental conditions in practice, we combined the number of the most suitable internal reference genes with geNorm and comprehensive rankings with RefFinder. As shown in Table 3, not only were the stable reference genes of different tissues different under the same hormone stress, but also the stable reference genes of the same tissue under different hormone stresses were different. For example, g20 and g23 (buds), g20, g22 and g10 (leaves), g26 and g12 (cambium regions), g20, and g12 and g10 (all samples) were the most stable reference genes in the respective sample sets under 6-BA stress. For buds, g20 and g23 (6-BA), g10 and g37 (ABA), g38 and g22 (BR), g39 and g10 (ETH), g29 and g22 (GA$_3$), g38 and g13 (IAA), g37 and g10 (MeJA) were the most stable reference genes under the respective hormone stresses. Under different hormone stresses, g10, g13 and g29 (buds), g23, g13 and g29 (leaves), g38, g39 and g23 (cambium regions), and g10, and g23 and g12 (all samples) were the most stable reference genes in the respective sample sets.

**Table 3.** The most stable reference genes under different hormone stress.

| Hormone | Tissue | | | |
| --- | --- | --- | --- | --- |
| | **Bud** | **Leaf** | **Cambium Region** | **Total** |
| 6-BA | g20, g23 | g20, g22, g10 | g26, g12 | g20, g12, g10 |
| ABA | g10, g37 | g23, g10, g24 | g20, g10 | g38, g20, g10 |
| BR | g38, g22 | g25, g42 | g9, g29 | g9, g38, g42 |
| ETH | g39, g10 | g38, g10 | g9, g20 | g20, g38, g10 |
| GA$_3$ | g29, g22 | g13, g6 | g38, g20 | g20, g38, g29 |
| IAA | g38, g13 | g7, g13 | g12, g20 | g7, g13, g10 |
| MeJA | g37, g10 | g38, g24 | g20, g22 | g38, g20, g10 |
| Total | g10, g13, g29 | g23, g13, g29 | g38, g39, g23 | g10, g23, g12 |

*3.4. Validation of Selected Reference Genes*

Expansin proteins are a class of specific proteins with plant cell wall extension ability [31]. Plant hormones and external stimuli (such as light, drought, hormones, salt stress, and hypoxia) affect the expression of expansin genes [32]. To demonstrate the usefulness of the best ranked candidate reference genes validated above, the expression patterns of *NcEXPA8* were analyzed under different hormone treatments [17]. According to the results of the RefFinder selection, the most stable reference gene, the most stable reference gene combination including two or three genes (Table 3), and the most unstable reference gene were used for normalization of the target gene. As shown in Figure 4, when one and the most stable reference gene combination were used for normalization respectively, *NcEXPA8* exhibited similar expression trends in a special tissue over time under a certain hormone treatment. However, when the most unstable reference gene was used for normalization, the expression profiles of *NcEXPA8* were quite different from that obtained using the stable reference genes.

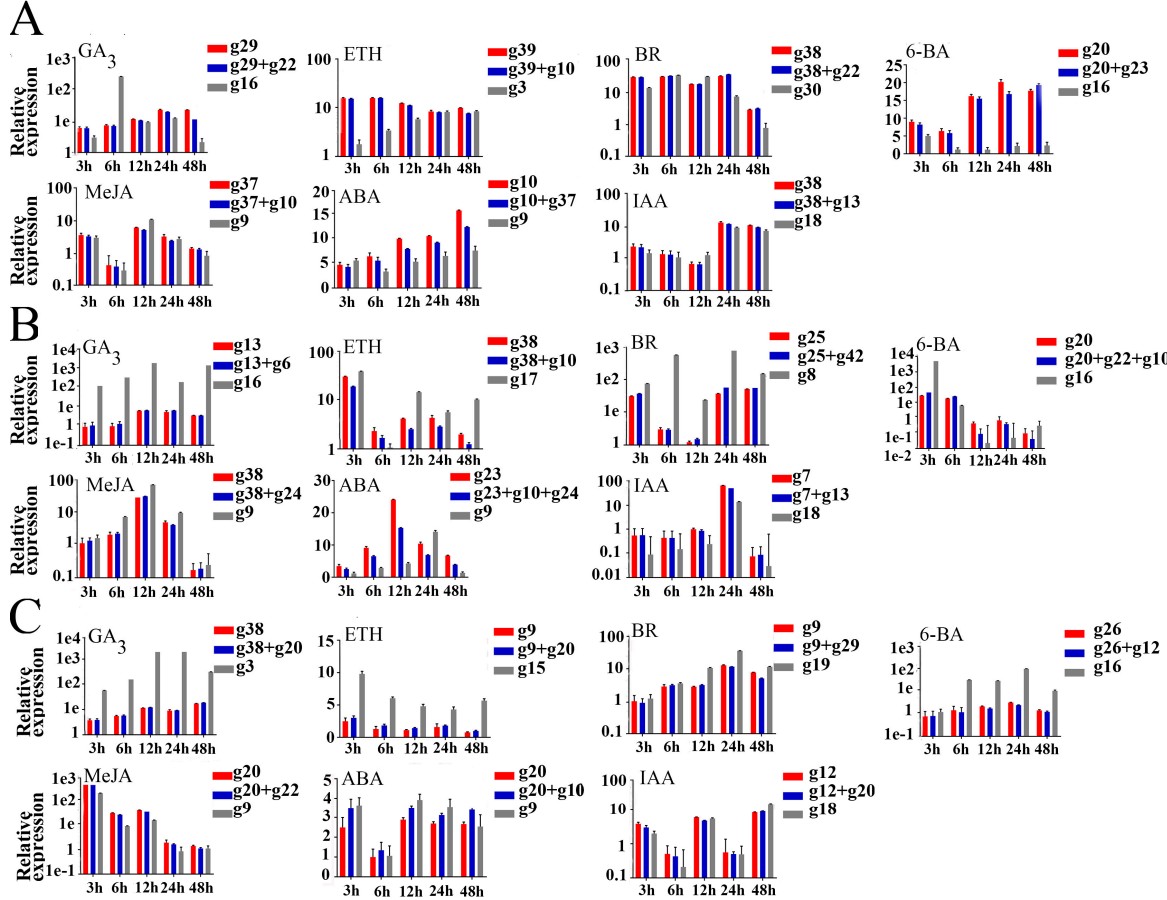

**Figure 4.** Relative quantification of *NcEXPA8* expression in buds (**A**), leaves (**B**), and cambium regions (**C**) under seven hormone treatments using different validated reference genes. The expression value of *NcEXPA8* was normalized with one or combinations of the most stable reference genes and the most unstable one. The expression level in each tissue treated after 0 h was set to 1. Each value represents the mean of three replicates, while vertical bars indicate the standard deviations (SD).

## 4. Discussion

Plant hormones are important regulators for plant growth and immunity. With the application of a plant model, particularly arabidopsis, many aspects of hormone biology have been elucidated. Most hormones are involved in many different processes during plant growth and development. This complexity is reflected in the contributions of hormone synthesis, transport, and signaling pathways, as well as the diversity of hormone interactions controlling the growth response [33,34]. In the past decade, there has been evidence that abscisic acid, gibberellin, cytokinin, auxin, and brassinoid steroids are associated with abiotic stress or developmental processes, and are key components of plant immune responses [35]. Many aspects of plant hormone signal transduction biology have been well characterized. Notably, receptors for nine plant hormones have been identified as intermediates between hormones and target genes [36]. Therefore, in order to reveal the target gene function, it is very important to study the expression level of target genes under hormone stress.

RT-qPCR has become a powerful tool for accurate gene expression analysis because of its high throughput, sensitivity, and accuracy [1–3]. However, several factors affect the quantification of gene expression, including the initial template amount, RNA quality, enzymatic efficiencies, and primer performance [4]. Stably expressed reference genes are the most commonly used to normalize RT-qPCR data, which can compensate for these variations [5]. The ideal reference genes should be expressed at a constant level across various conditions, such as developmental stages or tissue types. However, no one gene has an invariant expression under every experimental condition [7–10].

In addition, the traditional reference genes are not always expressed stably across species. Therefore, it is necessary to validate the expression stability of candidate reference genes under specific experimental conditions prior to their use for normalization, rather than using reference genes that have been published elsewhere [8,9]. Although stable internal reference genes have been obtained among different tissues of *N. cadamba* [7], there is no report on the selection of internal reference genes under hormone stresses, which is not conducive to seeking knowledge on the regulation and functions of key genes under hormone stresses.

*N. cadamba* is an important medicinal and afforestation tree. It grows rapidly and all tissues can be used as medicinal materials. Due to lack of effective genome information, the study of reference genes in *N. cadamba* has lagged behind that of other major plant species. We, thus, selected a series of candidate reference genes, the sequences of which could be obtained from our RNA-Seq database for *N. cadamba* stem [30]. In the present study, we developed a RT-qPCR method for 43 candidate reference genes belonging to 21 housekeeping gene families. It should be pointed out that although *18S rRNA* was frequently used as an internal control for normalization of RT-qPCR in earlier studies [37,38], it was not introduced in this study because it was not suitable for normalization of weakly expressed genes due to its very high expression level, with CT values of less than 15 across all samples in *N. cadamba*, even though the cDNA template was diluted 15-fold (data not shown).

The specificity of the primer pairs for the 43 candidate reference genes was confirmed by agarose gel electrophoresis (Figure 1A), melting curves analysis (Figure 1B), and sequencing of their amplicons. The expression stability of the candidate reference genes under different experimental conditions was ranked by RefFinder after evaluation using geNorm, NormFinder, ΔCt, and BestKeeper, respectively. All of the reference genes tested in the present study are members of gene families, some of which are large, except for *RAN*, with only one member (Supplementary S1). Therefore, it is difficult to obtain specific primers due to sequence similarity among members of one gene family. In the study, in order to ensure primer pair specificity, at least one primer in each primer pair was located in the 3′UTR of candidate reference gene, because the sequences of 3′UTR are more specific than that of ORF among the members of the same gene family [39].

When all *N. cadamba* samples were tested, g10 (*FPS1*) was overall the most stable and best candidate for the normalization of general gene expression for *N. cadamba*. However, most sets of samples had their own best reference genes (Table 2 and Tables S1–S32 of Supplementary S2). For instance, g20 (*PP2A*) ranked higher in the sets under GA$_3$, ETH, and 6-BA treatment; g38 (*UBCE*) under MeJA or ABA treatment; and g9 (*eIF*) and g7 (*EF1α*) under BR and IAA treatments, respectively. In addition, g10 (*FPS1*), g23 (*RPL*), and g38 (*UBCE*) ranked higher for the bud, leaf, and cambium region, respectively, under different hormone treatments. Furthermore, three tissue sets (bud, leaf, and cambium region) under the same hormone treatment, except for 6-BA, had their own best reference genes. For example, g29 (*SAMDC*), g13 (*FBK*), and g38 (*UBCE*) ranked higher in the bud, leaf, and cambium region, respectively, under GA$_3$ treatment, rather than g20 (*PP2A*), which was higher in all samples under the same treatment. This analysis indicated that the housekeeping genes are regulated differently in different tissues under different hormone treatments, and indicated the importance of reference gene validation for each experimental condition before their use for normalization in RT-qPCR, especially samples belonging to different sets.

Increasing evidence shows that no single gene can be used for accurate normalization in RT-qPCR data analysis and that normalization with two or more stable reference genes is preferred. We determined the optimum number of reference genes needed for accurate standardization. Although more reference genes for normalization will improve the accuracy of the result, this is expensive and time consuming in practice. Therefore, the number of reference genes should be taken into account. For example, the result of geNorm analysis showed that V$_{7/8}$ was slightly lower than 0.15 and V$_{6/7}$ was slightly higher than 0.15 in all samples under different hormone stresses (Figure 3), indicating that seven reference genes should be used together as the internal control under this specific experimental condition. However, this is not feasible in practice. Additionally, setting cutoff values for

geNorm was not necessary and at most three genes were sufficient to obtain more reliable normalization than a single reference gene [12]. Therefore, in this study, for the samples that required more than three reference genes for normalization using geNorm, only the three most stable reference genes were selected as the internal control (Figure 3, Table 3), according to the number of the most suitable internal reference genes using geNorm and comprehensive rankings with RefFinder.

The expression profiles of *NcEXPA8* in different tissues and certain functions are understood [17], although its expression profile under hormone treatment has not been reported. Furthermore, there have been no reports on the study of gene expression under hormone treatments or on gene function in *N. cadamba*, so the gene *NcEXPA8* was selected to validate reference genes. To illustrate the suitability of the reference genes validated in the study, the relative expression levels of *NcEXPA8* in all samples were compared with the best and worst candidate reference genes as controls for normalization. When the most unstable reference gene was used for normalization, the expression profiles of *NcEXPA8* were very different from or even opposite to that obtained using the most stable reference genes in a special tissue over time under a certain hormone treatment (Figure 4). More importantly, gibberelliin (GA), auxin (IAA), and methyl jasmonate (MeJA) responsive elements exist with TATA-boxes, TGA-elements, and CGTCA/TGACG-motifs, respectively, in the 2000 bp upstream region of *NcEXPA8* ORF (Supplementary S3). However, only under $GA_3$ treatment did the expression level of *NcEXPA8* show continuous upregulation over time, while under IAA/MeJA treatment its expression level fluctuated and was even downregulated at certain treatment time points (Figure 4), which was similar to the primary results reported in other gene expression studies under hormone treatments, showing downregualtion even though genes contained corresponding hormone-responsive elements [40]. As an important growth regulator, GA induces cell and stem growth through expansin-mediated loosening of the cell wall by increased expression and activity of expansins [41,42]. Additionally, overexpression of *NcEXPA8* resulted in longer fiber cells and higher stems [17], suggesting that GA might induce *NcEXPA8* expression and play an important role in *NcEXPA8* in *N. cadamba*. These results were consistent with the continuous upregulation of *NcEXPA8* expression over time with the stable reference genes for normalization, indirectly indicating the good stability of the selected reference genes. Therefore, all of these results suggested that stable reference genes are important for accurate quantification of target gene expression in *N. cadamba* under certain conditions.

## 5. Conclusions

This study screened the most stable internal reference genes under seven hormone treatments. The stability levels of internal reference genes were different under different hormone stresses. Additionally, among different tissues under the same hormone stress, the stability levels of reference genes were also different. This study also proved that no single gene was expressed stably in all tissue types or under all experimental conditions. However, the numbers of most suitable internal reference genes with geNorm and comprehensive rankings with RefFinder were taken into account together, showing that g10 (*FPS1*), g23 (*RPL*) and g12 (*FBK*) were the most stable reference genes in all samples, which would be used as internal reference genes together for normalization.

**Supplementary Materials:** The following are available online at http://www.mdpi.com/1999-4907/11/9/1014/s1: Supplementary S1: The housekeeping gene families and corresponding unigenes in *N cadamba*. Supplementary S2: Stability evaluation of reference genes in different tissues under different hormone stresses. Supplementary S3: The 2000 bp upstream region of NcEXPA8 ORF and cis-regulatory element analysis with the PlantCARE online service (http://bioinformatics.psb.ugent.be/webtools/plantcare/html/).

**Author Contributions:** Conceptualization, X.C. and K.O.; data curation, J.L. and C.L.; methodology, B.L.; writing—original draft, D.Z.; writing—review and editing, X.C. All authors have read and agreed to the published version of the manuscript.

**Funding:** This study was funded by Science and Technology Program of Guangdong, China (2017B020201008); Natural Science Foundation of China (31600525); Forestry, Science, and Technology Innovation Project in Guangdong Province (2018KJCX001, 2019KJCX001); Extension and Demonstration Project of Forestry, Science,

and Technology from the Chinese state financial budget ((2018)GDTK-08); and the Science and Technology Program of Guangzhou (201904020014).

**Conflicts of Interest:** The authors declare no conflict of interest.

## Abbreviations

| | |
|---|---|
| 6-BA | 6-Benzylaminopurine |
| ΔCt | Delta cycle threshold |
| ABA | Abscisic acid |
| *ACT* | Actin |
| APT | Adenine phosphoribosyl transferase |
| BR | Brassinolide |
| *CAC* | Clathrin adaptor complex medium |
| CTAB | Cetyltrimethylammonium bromide |
| *CYP* | Cyclophilin |
| DEPC | Diethyl pyrocarbonate |
| *EF1α* | Elongation factor 1α |
| *eIF* | Eukaryotic initiation factor |
| ETH | Ethephon |
| *FBK* | F-Boxkelch repeat protein |
| *FPS1* | Farnesyl pyrophosphate synthase 1 |
| GA | Gibberellic acid |
| *GAPDH* | Glyceraldehyde-3-phosphate dehydrogenase |
| IAA | Indole-3-acetic acid |
| MeJA | Methyl Jasmonate |
| NCBI | National Center for Biotechnology Information |
| *PEPKR1* | Phosphoenolpyruvate carboxylase-related kinase 1 |
| *PP2 A* | Protein phosphatase 2 A |
| *RAN* | GTP-binding nuclear protein |
| *RPL* | Ribosomal protein L |
| *RPS* | Ribosomal protein S |
| *RuBP* | Ribulose 1,5-bisphosphate carboxylase |
| *SAMDC* | S-adenosylmethionine decarboxylase |
| *TEF* | Translation elongation factor |
| *Tub-α* | Tubulin α |
| *Tub-β* | Tubulin β |
| *UBCE* | Ubiquitin conjugating enzyme |
| *UBQ* | Ubiquitin |
| *UPL* | Ubiquitin protein ligase |

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
