# Peer review of "Internal Reference Gene Selection under Different Hormone Stresses in Multipurpose Timber Yielding Tree Neolamarckia cadamba"

_forests, doi:10.3390/f11091014_

Round 1

Reviewer 1 Report

The authors describe a set of Reference genes, which can be used to analyze gene expression in response to hormonal treatment in Neolamarckia cadamba. The analysis used to identify this set of genes is well explained and it will be useful for researchers working on this specie.  

However, I had some concerns about Figure 4:

  1. It is not very readable. Legends should be bigger. I would suggest splitting the figure, to have the gene expression of bud, leaf and cambium in three separated figures.
  2. Considering that the time-point 0 is set to 1, NcEXPA8 seems to be induced by any of the hormones tested, which is particularly surprising in the case of ABA. At line 256-258, the authors claim that the expression on EXPA8 has been tested in response to hormone treatment, however, I could not find experiments showing this in the paper cited. If the induction of EXPA8 has been clearly shown in the past, authors should cite the right literature. On the contrary, if this is new evidence, they should comment it.
  3. Related to point 2, the presence of the control sample at 0h, or even better the samples treated with the diluent of the hormone, should be included in the figure for a better interpretation of the results.
  4. Line 263-264. The authors say that the expression of EXPA8 is different if an unstable reference gene was used for normalization in all the samples. However, MeJA and IAA the expression pattern seems to be similar to all the three combinations of reference genes used. How do authors comment on this? If they observed some differences, I think that they should provide statistical analysis.

Minor comments:

  1. Given that the paper is focused on gene expression, I think that authors should provide more details about RNA isolation protocol for the reproducibility of the results.
  2. Line 113 of Materials and Methods. The time-points of hormone treatments don’t match with the figure. 3h is missing.

Author Response

Point 1: 
It is not very readable. Legends should be bigger. I would suggest splitting the figure, to have the gene expression of bud, leaf and cambium in three separated figures

Response 1: As a whole, the purpose is to illustrate the importance of using stable internal reference genes in quantitative analysis of gene expression level, so it is not suitable to split. But modify the legend by replacing "validated reference genes" with "different validated reference genes (s)" in Line 271.

Point 2: Considering that the time-point 0 is set to 1, NcEXPA8 seems to be induced by any of the hormones tested, which is particularly surprising in the case of ABA. At line 256-258, the authors claim that the expression on EXPA8 has been tested in response to hormone treatment, however, I could not find experiments showing this in the paper cited. If the induction of EXPA8 has been clearly shown in the past, authors should cite the right literature. On the contrary, if this is new evidence, they should comment it.

Response 2: It has been shown that expression of some expansin genes(EXPs) often were induced by hormones in the paper cited (Reference 32). Though the expression of NcEXPs under different hormone stress has not been reported, the function of NcEXPA8 has been understood to a certain extent in the Reference 17. In this article, our main concern is the difference of the expression of NcEXPA8 between the stable internal reference gene(s) and the unstable internal reference gene, so we do not comment the expression of NcEXPA8 in the article.

Point 3: Related to point 2, the presence of the control sample at 0h, or even better the samples treated with the diluent of the hormone, should be included in the figure for a better interpretation of the results.

Response 3: Because we consider the expression at 0h as 1, we did not put it in the figure. Moreover our focus is not on the detail expression level of NcEXPA8 from 0h to 48h, but on the expression trend of NcEXPA8.

Point 4: Line 263-264. The authors say that the expression of EXPA8 is different if an unstable reference gene was used for normalization in all the samples. However, MeJA and IAA the expression pattern seems to be similar to all the three combinations of reference genes used. How do authors comment on this? If they observed some differences, I think that they should provide statistical analysis.

Response 4: Under MeJA and IAA stress, the expression pattern of NcEXPA8 still have difference. Such as MeJA(c: from 24h to 48h) and IAA (a: from 6h to 12h), when the stable reference gene was used that the expression of NcEXPA8 is down-regulation, and when the unstable reference gene was used that the expression of EXPA8 is up-regulation.

Point 5: Given that the paper is focused on gene expression, I think that authors should provide more details about RNA isolation protocol for the reproducibility of the results.

Response 5: The method of RNA isolation from different tissues of N. cadamba is perfect and has been widely used in our laboratory(Reference 29 and 30 in the original manuscript). So the reference on the RNA isolation method was  directly into “Materials and methods” and RNA isolation from different tissues of N. cadameba followed this method exactly, but the steps of RNA extraction are not described in detail.

Point 6:  Line 113 of Materials and Methods. The time-points of hormone treatments don’t match with the figure. 3h is missing.

Response 6: I am so sorry about that the time point 3h after hormone stresses was missed in the original manuscript in Line 114 due to my carelessness. So the time point 3h have been added in Line 114.

Reviewer 2 Report

In this manuscript, Zhang et al provide a systematic analysis of screening reference genes for qRT-PCR experiments in the tree species Neolamarckia cadamba. In general, I think this study is well designed. The presentation is clear. Their results suggested that among 43 selected housekeeping genes, none of them showed a constant expression level across samples collected from different stress conditions. The authors, then evaluated the reliability of selected genes under different conditions using different algorithms. They identified several stable genes for each condition, and validated the expression of these genes using NcEXPA8, a well-known gene. This will be very useful for research community working on the species Neolamarckia cadamba.

However, I think, there are some points that the authors could consider to improve the manuscript.

Please find my comments below:

  1. Throughout the manuscript, the use of qRT and RT-qPCR should be consistent.
  2. Supplementary A2 would be easily navigated if the authors present it in an Excel file containing different worksheets corresponding to the number of tables they have.
  3. Line 113: which part of the plant the cambium regions were collected from? This should be mentioned somewhere in the manuscript. The authors could consider providing some explanation why cambium sample is selected for this analysis?
  4. It would also good to add a brief description of how cambium samples were collected, even the authors already cited the original paper.
  5. Reference sources for 23 housekeeping genes in Arabidopsis should be added to the sentence in line 118.
  6. Line 167: what is the “stem database” referring to?
  7. Line 167: please add a reference to the program Primer5.
  8. Line 174: was it Sanger sequencing?
  9. In Figure 1, please increase the font size for the labels, and add labels to x axis (melting curves).
  10. Line 185: it would be helpful to indicate how many samples were used, corresponding to “under different hormone stresses”.
  11. Line 187: I wonder if one can tell the expression level of each genes based on the PCR band intensity in Figure 1A. In this case, using leaf cDNA, g7 and g25 had really low expression level (much lower than g8, which had the lowest mean Ct). Were these consistent with the Ct values? Or was panel 1A used only for checking product size without controlling the loaded volume?
  12. Figure 2: would that be more informative to sort the data from low to high? I understand that the authors might just want to show the gene names in the order, but I think it might be nice to sort it, as here we also want to see the min and max values and the range.
  13. Also in Figure 2 legend or in text, the number of samples (data points) used to generate the boxplots should be mentioned.
  14. For the validation of NcEXPA8 expression using the most/ least stable reference gene candidate, how did the authors know that “NcEXPA8 exhibited similar expression trends under each hormone treatment” truly reflects the in vivo expression trends of the gene? I think the authors should describe the expected expression level of NcEXPA8 in each tissue tested based on published works, and then compare against their results normalized against the most stable and least stable reference genes. In the current version, this is not clear to me.
  15. In the reference #17 (cited in this manuscript), the genes Cyclophilin (JX902587) was used as internal reference gene. Did the authors compare their results with this gene?

Author Response

Point 1: 
Throughout the manuscript, the use of qRT and RT-qPCR should be consistent

Response 1: “RT-qPCR”, abbreviation of “Real time quantitative PCR”, was used throughout the original manuscript.  “qRT-PCR” was used Only in Line 407 in Reference.

Point 2: Supplementary A2 would be easily navigated if the authors present it in an Excel file containing different worksheets corresponding to the number of tables they have

Response 2: Supplementary A2 has been replaced with an Excel file containing the same worksheets and reuploaded.

Point 3: Line 113: which part of the plant the cambium regions were collected from? This should be mentioned somewhere in the manuscript. The authors could consider providing some explanation why cambium sample is selected for this analysis?

Response 3: “cambium regions” was collected according to the Reference 29. So in order to save space, no specific operation of cambium regions acquisition was provided. Because of the common sense of that cambium plays an important role in second growth of tree trunk, the cambium sample was selected for this analysis. Addationally, the other two tissues are very important in plant growth and develop, these are also common senses. Furthermore, the explanation about the other tissues selection should be added if providing some explanation about cambium selection, which will take up too much space. So the explanation was not provided.

Point 4: It would also good to add a brief description of how cambium samples were collected, even the authors already cited the original paper.

Response 4: The method of cambium sample collection is adopted extensively according to previous description and this will take up some space. So no specific operation of cambium regions acquisition was provided.

Point 5: Reference sources for 23 housekeeping genes in Arabidopsis should be added to the sentence in line 118

Response 5: The 23 housekeeping genes were determined according to previous report described in the second paragraph in Introduction and the corresponding amino acid sequences were downloaded from the A. thaliana TAIR10 database. So the determination of .housekeeping genes was not described in “Materials and method”.

Point 6: Line 167: what is the “stem database” referring to?

Response 6: Transcriptome data were obtained from previous stem segment samples, so the database is referred to as “stem database”.

Point 7: Line 167: please add a reference to the program Primer5.

Response 7: The details “(Premier Biosoft Interpairs, Palo Alto,CA)” were inserted just after the word “Primer5” in Line 168.

Point 8: Line 174: was it Sanger sequencing?

Response 8: Yes, the fragment was sequenced with Sanger.

Point 9: In Figure 1, please increase the font size for the labels, and add labels to x axis (melting curves).

Response 9: The Figure 1 was modified and re-insert the corresponding position in the manuscript. The font size for the labels has been increased in Figure 1A. X axis was added label with the temperature unit °C in original manuscript.

Point 10: Line 185: it would be helpful to indicate how many samples were used, corresponding to “under different hormone stresses”.

Response 10: Thanks very much. It is helpful to indicate number ofthe used samples under each hormone stresses. So, the sentence “The CT value for each gene was from 324 samples (6 time-points per tissue set and 3 tissue sets per hormone stress)” has been inserted in Line 186.

Point 11: Line 187: I wonder if one can tell the expression level of each genes based on the PCR band intensity in Figure 1A. In this case, using leaf cDNA, g7 and g25 had really low expression level (much lower than g8, which had the lowest mean Ct). Were these consistent with the Ct values? Or was panel 1A used only for checking product size without controlling the loaded volume?

Response 11: The main purpose of Figure 1A was to check product size and primer specificity. Additionally, it could also indirectly reflect the expression of genes in untreated leaves to a certain extent, because the number of templates added in PCR and the loaded volume for each gene is basically the same. This can only explain the specific situation of leaves without treatment, but neither  represent the expression under hormone treatment, nor reflect the expression of genes in other tissues. Therefore, it is normal that the results in Figure 1A are different from the average CT values of all samples for one gene.

Point 12: Figure 2: would that be more informative to sort the data from low to high? I understand that the authors might just want to show the gene names in the order, but I think it might be nice to sort it, as here we also want to see the min and max values and the range.

Response 12: Yes, we hoped that figure 2 can show both the average expression level and the variation of expression level for one gene across all samples. Even if the classification is based on the span, it is not easy to see how much the specific span is, because the min/max CT value of each gene is inconsistent, and the picture is a bit messy, because the gene names on the abscissa are not arranged in order.

Point 13: Also in Figure 2 legend or in text, the number of samples (data points) used to generate the boxplots should be mentioned.

Response 13: The number of samples used to generate the boxplots has been inserted in the Line 186 with the sentence “The CT value for each gene was from 324 samples (6 time-points per tissue set and 3 tissue sets per hormone stress)”.

Point 14: For the validation of NcEXPA8 expression using the most/ least stable reference gene candidate, how did the authors know that “NcEXPA8 exhibited similar expression trends under each hormone treatment” truly reflects the in vivo expression trends of the gene? I think the authors should describe the expected expression level of NcEXPA8 in each tissue tested based on published works, and then compare against their results normalized against the most stable and least stable reference genes. In the current version, this is not clear to me.

Response 14: Thanks very much for your precious comment. Yes, what we want to express is that when one and the most stable reference gene combination were used for normalization respectively, NcEXPA8 exhibited similar expression trends in certain tissues over time under certain hormone treatment. So, the sentence “when one or the most stable reference gene combination were used for normalization, NcEXPA8 exhibited similar expression trends under each hormone treatment.” in Line 266-Line 268 has been replaced with the sentence “when one and the most stable reference gene combination were used for normalization respectively, NcEXPA8 exhibited similar expression trends in a special tissue over time under a certain hormone treatment”. And “ in a special tissue over time under a certain hormone treatment” was inserted just after “the most stable reference genes” in Line 359.

Point 15: In the reference #17 (cited in this manuscript), the genes Cyclophilin (JX902587) was used as internal reference gene. Did the authors compare their results with this gene?

Response 15: The gene Cyclophilin (JX902587) was used as internal reference gene for NcEXPA8 expression analysis in different growth tissue without any treatment in the Reference 17. And we also compared Cyclophilin (JX902587) with the selected internal reference genes in different tissues of N. cadamba (Zhang Deng, Li J.J.; Zhang M.J.; Bao Y.T.; Yang X.; Xu W.Y.; Ouyang K.X; Chen X.Y. Selection and Validation of Reference Genes for Quantitative RT-PCR Analysis in Neolamarckia cadamba. Chinese Bulletin of Botany, 2018, 53(6): 829-839.). We found that its stability is not as good as internal reference genes selected in that paper, so we did not compare with it in this article.

Round 2

Reviewer 1 Report

I agree that the focus of the paper is not the expression of EXP8, but since the authors use this gene to validate their reference genes, the EXP8 expression data should be reliable. Since in the references cited there are no specific data about the expression of EXP8 in response to all these hormones in the conditions/tissues used in this paper and these hormones have different effects on plant growth, I think the authors should have discussed the EXP8 expression trend more extensively. Or chose different genes for the different hormones treatment.

I still think that Figure 4 is hard to read and to interpret and it should be improved.

Lastly, providing the details of the materials and methods used in the paper is important for the reproducibility of the data. Even if the protocol has been used in the past, it is very useful having the whole protocol in the paper, especially if it is an important step of the experiments shown in the paper, as the RNA extraction is in this one.

Author Response

Point 1: I agree that the focus of the paper is not the expression of EXP8, but since the authors use this gene to validate their reference genes, the EXP8 expression data should be reliable. Since in the references cited there are no specific data about the expression of EXP8 in response to all these hormones in the conditions/tissues used in this paper and these hormones have different effects on plant growth, I think the authors should have discussed the EXP8 expression trend more extensively. Or chose different genes for the different hormones treatment.

Response 1: Thanks for your suggestion. The gene NcEXPA8 was selected to validate reference genes, because its expression profile in different tissues and certain functions have been understood although its expression profile under different hormone treatments was not reported. Furthermore, there were no reports about study of gene expression under hormone treatment or function in N. cadamba. So in the last paragraph of Discussion, we have added a description of that why the gene NcEXPA8 was selected to validate reference genes and discussed the expression trend more extensive. At the beginning of the paragraph, we have added the sentences “Because NcEXPA8 expression profile in different tissues and certain functions have been understood although its expression profile under hormone treatment was not reported. Furthermore, there were no report about study of gene expression under hormone treatment or function in N. cadamba, so the gene NcEXPA8 was selected to validate reference genes.” and the description of expression trend with “More importantly, there are gibberelliin (GA), auxin (IAA) and methyl jasmonate (MeJA) responsive element with TATA-boxes, TGA-elements and CGTCA/TGACG-motifs respectively in 2000 bp upstream region of NcEXPA8 ORF (Supplementary A3). However, only under GA3 treatment, the expression level of NcEXPA8 showed continuous up-regulation over time, while under IAA/ MeJA treatment, its expression level fluctuated and was even down-regulated at certain treatment time points (Figure 4), which was similar to the primary reported in other gene expression under hormone treatments showing down-regualtion even though contain corresponding hormone responsive elements [40]. As an important growth regulator, GA induces cell and stem growth through expansin-mediated loosening of the cell wall by increased expression and activity of expansions [41,42]. Additionally, overexpression of NcEXPA8 resulted in longer fiber cells and higher stem [17], suggesting that GA might induce NcEXPA8 expression and play an important role of NcEXPA8 in N. cadamba. These was consistent with the contimuous up-regulation of NcEXPA8 expression over time with the stable reference gene(s) for normalization, indirectly indicated good stability of the selected reference gene(s).”.

Point 2: I still think that Figure 4 is hard to read and to interpret and it should be improved.

Response 1: The Figure 4 was splitted to 3 figures according to three sample sets. And the two sentences of primary figure legend were replaced with “Relative quantification of NcEXPA8 expression in buds (A), leaves (B) and cambium regions (C) under seven hormone treatment using different validated reference gene(s).”

Point 3: Lastly, providing the details of the materials and methods used in the paper is important for the reproducibility of the data. Even if the protocol has been used in the past, it is very useful having the whole protocol in the paper, especially if it is an important step of the experiments shown in the paper, as the RNA extraction is in this one.

Response 3: The details of RNA extraction were provided with “Frozen tissue was ground to a fine powder in liquid nitrogen, using a mortar and pestle. Then, 100 mg samples of the powder were transferred into individual 1.5 mL RNase-free tubes containing 600 mL of prewarmed extraction buffer at 60 C. The extraction buffer contained the following: 2% CTAB, 2% polyvinylpyrrolidone (PVP) K-40, 100 mmol/L TrisHCl (pH 8.0), 25 mmol/L ethylenediaminetetraacetic acid(EDTA; pH 8.0), 2.0 mol/L NaCl, 2 g/L spermidine and 2% b-mercaptoethanol (added immediately before use). The extracts were mixed by vortexing and incubated at 60 °C in a water bath for 10 min with vigorous shaking for several times. An equal volume of chloroform/isoamyl alcohol (24:1) was added to the homogenate and was mixed completely by vortexing. The mixture was centrifuged at 12,000 r/min for 10 min at 4 °C, except for bud samples, which were centrifuged for 20 min. The supernatant was transferred to a new tube and the above step was repeated. The supernatant was then transferred to a new tube and the next operation was carried out according to the instructions of the Plant RNA Kit (R6827, OMEGA).” In Line 128.

Reviewer 2 Report

In this version, the authors have addressed some of my comments. For other comments, they provided the explanations for not revising it.

Author Response

There were no comment from reviewer 2, thank you!

This manuscript is a resubmission of an earlier submission. The following is a list of the peer review reports and author responses from that submission.

Round 1

Reviewer 1 Report

The authors have tested several putative reference genes for qPCR analyses of Neolamarckia cadamba by using four commercial or free softwares. The results of the study are species-specific and do not represent novel findings for broader audience. Therefore, I do not recommend publication of the manuscript in its current form.

Specific comments for the authors:

Hormonal treatments are very central part of the study. However, it is not explained how they were carried out. How were the reference genes chosen for the study? The text, tables and figures are very difficult to read because the gene IDs are only specified in the supplementary table 1. Figure 2: What is the meaning of the gel picture? List of abbreviations: Please remove the website address.

Reviewer 2 Report

The manuscript by Zhang et al., reports the reference genes to be used in qPCR studies on N. cadamba. The study is well designed, data is systemically presented and manuscript is clearly written. I have only one comment that authors can address to increase the impact of their work. The shortlisted candidate genes (FPS1, RPL and FBK) have been tested only on N. cadamba in this study. These genes can be tested on other members of  Rubiaceae family; a simple experiment without involving hormone stresses. This small experiment can answer important question of whether the reported genes can (or can not) be used by the researchers working on related tree species.